# Validating the Use of Gaussian Process Regression for Adaptive Mapping of Residual Stress Fields

**DOI:** 10.3390/ma16103854

**Published:** 2023-05-20

**Authors:** Chris M. Fancher, Singanallur Venkatakrishnan, Thomas Feldhausen, Kyle Saleeby, Alex Plotkowski

**Affiliations:** 1Material Science and Technology Division, Oak Ridge National Laboratory, Oak Ridge, TN 37831, USA; plotkowskiaj@ornl.gov; 2Electrification and Energy Infrastructure Division, Oak Ridge National Laboratory, Oak Ridge, TN 37831, USA; 3Manufacturing Science Division, Oak Ridge National Laboratory, Oak Ridge, TN 37831, USA

**Keywords:** neutron stress mapping, autonomous experiments, engineering

## Abstract

Probing the stress state using a high density of measurement points is time intensive and presents a limitation for what is experimentally feasible. Alternatively, individual strain fields used for determining stresses can be reconstructed from a subset of points using a Gaussian process regression (GPR). Results presented in this paper evidence that determining stresses from reconstructed strain fields is a viable approach for reducing the number of measurements needed to fully sample a component’s stress state. The approach was demonstrated by reconstructing the stress fields in wire-arc additively manufactured walls fabricated using either a mild steel or low-temperature transition feedstock. Effects of errors in individual GP reconstructed strain maps and how these errors propagate to the final stress maps were assessed. Implications of the initial sampling approach and how localized strains affect convergence are explored to give guidance on how best to implement a dynamic sampling experiment.

## 1. Introduction

Residual stresses are persistent stresses in a material that can arise during manufacturing or while a component is in service [1]. While residual stresses can affect the performance and lifetime of a component through premature failure or cracking that limits use, designed compressive stress states can improve component lifetimes [2]. Often stresses are unavoidably introduced during fabrication and joining, where thermal gradients introduce stresses via differential thermal expansion. The advancement of additive manufacturing underscores the need to develop new approaches for mitigating how stresses evolve [3]. For example, applying lessons designed for controlling the stress state during welding, such as low-temperature transformation materials, has shown promise for tuning stresses in wire-arc additively printed components [4]. Understanding how stresses evolve with changes in build strategies and materials is critical for developing new approaches for controlling and limiting stress-induced failures.

Neutron diffraction is an ideal characterization method for understanding how processing and joining approaches affect the residual stress state [5]. Engineering neutron diffractometers are well suited for determining the three-dimensional stress state of a material. These instruments utilize optics that enable the measurement of diffraction data from localized 1–25 mm^3^ volumes. Translating components through this volume allows the non-destructive measurement of strain distributions. While the weak interaction of neutrons is advantageous for probing bulk stresses buried in a sample, these measurements require long count times, and sampling large components is time intensive. In general, determining the principal stress state in a material involves measuring strains along three orthogonal directions [1]. Strain maps are typically measured using a one or two dimensional grid of strain mapping locations. For example, a single one-dimensional line of data is needed to determine the stress state in a welded plate [6]. The layer-by-layer nature of additive manufacturing results in a thermal history that can introduce a more complex stress state [4,5]. These complexities often necessitate the measurement of higher dimensional maps to capture underlying variations in the stress state [7].

While higher dimensional strain grids offer more significant insights into the spatial variations in the strain distribution, these maps are far more time insensitive. For example, measuring stresses in a thin-walled component with dimensions 50 mm × 100 mm using a 5 mm × 5 mm measurement grid would require 171 measurements per strain direction, assuming a 5 mm offset to mitigate partially buried gauge volumes at the edges (45 mm × 95 mm sampled using 5 mm × 5 mm). A strongly scattering material with a large gauge volume might require 30 s/point, or ~1.5 h/direction, and ~4.5 h to complete all three directions. Decreasing the size of the gauge volume or probing a weak scattering material can increase the count time to 5 min or more per point or ~19 h. A finer grid spacing, e.g., a 2.5 mm × 2.5 mm grid, would increase the sample to ~7 h and 72 h for single point measurement times of 30 s and 5 min, respectively. Higher fidelity 2D strain data, and those with a finer sampling grid, are useful for quantifying materials’ strain/stress state. However, the prolonged measurement times limit their viability, because access to engineering diffractometers is highly constrained.

Recently, the authors proposed using Bayesian optimization and Gaussian process regression (GPR) as a novel approach for reducing the number of independent data measurements needed to reconstruct 2D strain fields [8]. The core idea behind the system was a *measure–infer–predict* loop to sequentially make measurements such that important regions (defined in some manner) are sampled first, thereby leading to a better overall view of the strains in the sample, compared to a sequential point-wise raster scanning approach. A wire-arc additively manufactured (WAAM) wall of a low-temperature transition (LTT) alloy was used as a demonstration, where the 2D strain field along the welding direction. These results evidenced that a GPR approach can reconstruct the full 2D strain distribution from a subset of measurement points and suggested that a GPR-driven experiment can reduce measurement times by 1/3 to 1/2. However, the work was limited to a single strain direction. Further investigation is needed to validate that a GPR approach is viable for mapping stresses, requiring measurement of multiple orthogonal strain directions and subsequent calculation of the stress state. Features within the strain field in each direction can require a different subset of measurement locations to obtain the best reconstruction.

In this work, we determine the feasibility of using the GPR approach to reduce the number of independent measurements needed to reconstruct the 2D stress distributions in WAAM walls. Uncertainty in the reconstructed strain fields was assessed by masking points from the GPR sample and determining the difference between the recovered and measured data. Effects of how these differences propagate to uncertainties in the resulting residual stresses are explored. Results presented evidence that calculated residual stresses using a reconstructed strain field are valid, and resulting stresses are in excellent agreement with stresses calculated from full strain maps, demonstrating that a GPR-based autonomous experiment is a feasible approach for reducing the number of data points needed to capture the stress state of a material.

## 2. Materials and Methods

### 2.1. Sample Preparation

Planar walls were manufactured via WAAM using a custom system comprised of a Tormach ZA6 robotic arm and a Lincoln Electric R450 MIG welder. Two steel alloys, mild steel (MS), and a low-temperature transformation (LTT) steel [9], were used. The two alloys were selected to produce different residual stress distributions, as the austenite to martensite transformation in the LTT steel is known to modify the stress state due to the volume difference between the two phases [4]. The differences in the complexity of the resulting stress patterns provide a useful test case for the GPR approach. The toolpath for WAAM fabrication was generated using hyperMILL developed by OPEN MIND, and post-processed using Eureka from Roboris. The slicing strategy was comprised of traditional longitudinal raster patterns. While the same number of layers and the same scan pattern were used for both steels, the differences in material properties and process parameters resulted in different layer heights, meaning that the total build height was different, with the LTT sample tending to be taller, as shown in Figure 1. Processing parameters for both alloys are shown in Table 1. The samples were deposited on mild steel substrates prepared by machining and cleaned with alcohol. The wire diameter for both alloys was 1.14 mm.

### 2.2. Neutron Stress Mapping

Neutron diffraction data were measured at the High Flux Isotope Reactor using the high-intensity diffractometer for residual stress analysis [10,11]. Neutron optics defined a nominally 5 × 5 × 5 mm^3^ gauge volume (GV). Spatially dependent diffraction data were measured in a 5 × 5 mm grid to probe the two dimensional lattice strain distribution, with the GV centered along the depth. Bulk principal stresses were studied by measuring strain distributions for scattering vectors aligned along the weld (11), transverse (22), and build (33) directions. The 211 reflection peak was used as the 211 lattice stresses represent the bulk state [1].

Measured diffraction data were reduced from neutron events into 1-dimensional intensity vs. 2*θ* vectors and then analyzed via single peak fitting to extract site-specific interatomic spacing using the pyRS software package [12]. A pseudo-Voit peak shape function with a linear background was used to model the measured diffraction data. Interatomic spacing was determined using the following:(1)$$d_{hkl}(x,y,z)=\frac{\lambda }{2sin\theta (x,y,z)}$$
where *λ* is the neutron wavelength, and *θ* is the peak center extracted via single peak fitting. Lattice strains are determined from measured interatomic spacing through the following:(2)$$\varepsilon_{ii}(x,y,z)=\frac{d_{hkl}^{ii}(x,y,z)}{d^{0}}-1$$
where *d*^0^ is the stress-free reference lattice. The stress-free reference sample in this work was obtained through a stress-relief heat treatment by isothermal holding at 700 °C for 12 h. Stress relief was validated by ensuring that stress-free lattice parameters for the 11 (d_211_ = 1.0470(1) Å) and 22 (d_211_ = 1.0471(3) Å) scattering vectors were equivalent within estimated experimental errors.

Principal residual stresses were determined from the measured elastic strains using the following:(3)$$\sigma_{ii}=\frac{E}{\left(1+v\right)}\left[\varepsilon_{ii}+\frac{v}{1-2\nu }\left(\varepsilon_{11}+\varepsilon_{22}+\varepsilon_{33}\right)\right]$$
where *E* and *ν* are Young’s modulus and Poisson’s ratio, respectively.

### 2.3. Gaussian Process Regression Mapping

This work uses the GPR approach presented in [8] to reconstruct the strain fields from a subset of experimental data points. GP has become a popular machine-learning technique that can infer values of unknown points on a grid from a partial set of measurements [13]. For brevity, details of GP are not presented here, and readers are referred to ref [13] for greater detail. The GP approach in this work implements a Bayesian optimization to infer what point that has the highest variance using the GPim library [14]. The GPim library is available at https://github.com/ziatdinovmax/GPim. See ref. [14] for details of the implemented methods.

Measured neutron strain maps were reconstructed using independent GPRs. The Bayesian Optimization and GPR were retroactively applied to fully measured strain maps to evaluate the method’s accuracy based on partial scans. An initial set of grid points was used to initialize the GP method, after which the sequential scanning method was applied. Strain maps for the LTT and MS contained 11 × 14 and 11 × 12 (x, z) grid points, respectively. Data were transformed from the measured interatomic spacing (d_211_(x, y, z)~1.0470 Å) to a [0, 1] interval for ease of implementation. For simplicity, data were normalized using the min and max of the measured data from the corresponding strain map. Reconstructed maps were transformed back to DSpace for subsequent strain/stress analysis.

The overall performance of the GPR model was quantified using a modified K-fold cross-validation procedure. A random set of points (10%) were masked and excluded from the GPR training set, and the performance of the GPR was assessed based on the predictions for these points compared against the withheld data. For consistency, masked points were held constant for all strain directions. Differences between the measured and recovered data for the masked locations were calculated. These differences were evaluated for raw interatomic spacing, lattice strain, and residual stress. A set of ten trials (k=10) was used to test the error variability. Each GPR was initialized using 16 and 14 randomly selected non-masked locations for the LTT and MS, respectively. Effects of the initialization approach were assessed by comparing how structured vs. random initial point selection strategies and the fraction of initialization points included impacted the convergence of the reconstructed strain maps. Structured grids were devised to select points evenly spaced throughout the wall. The random initialization grid was constructed by iteratively selecting pixel ids from a list of non-masked pixel ids via the numpy.random.randint function. Selected pixel ids were removed from the selection list to ensure a pixel id could only be selected once. Pixel ids were selected until sufficient points were identified to achieve the desired initialization fraction. Appendix A visualizes three random initialization grids to illustrate how the grids differ.

## 3. Results

### 3.1. Measured Residual Strain Maps

Validating a novel approach for strain mapping requires knowledge of the expected strains. Stresses in a material are balanced by the forces along orthogonal directions, where the principal strains represent the balancing forces. Figure 2 summarizes the 2D strain maps and shows different strain patterns for the MS and LLT walls. Strains along each principal direction (11, 22, and 33) have unique distributions. For example, mild steel has a relatively smooth distribution, while LTT has sharp gradients near the sample base. The presence of significant strain gradients can increase the uncertainty in measured data and offers a route to determine if the GP approach is robust for capturing strain gradients and how strain gradients affect convergence.

### 3.2. Validation of GPR for Reconstructing Residual Stress Maps

The HIDRA instrument at ORNL measures each strain direction independently, and therefore, GPR was implemented to reconstruct a strain field without information from other strain directions. This approach is validated by masking 10% of the total measurement points and determining if the resulting reconstructed strain fields recover these masked points. A set of ten random trials were used to determine a distribution of errors.

In the first step, the GPR must reconstruct the measured distribution in interatomic spacing; otherwise, the resulting strains would not represent the true distributions within the material. To assess this requirement, the average normalized deviation between the measured and reconstructed interatomic spacing was used as a metric:(4)$$\frac{\sum^{N}\frac{d_{211}^{ii,R}-d_{211}^{ii,M}}{d_{211}^{ii,M}}}{N}$$
where *N* is the total number of masked points over the ten random trials, and $d_{211}^{ii,R,M}$ are the reconstructed (*R*) and measured (*M*) interatomic spacing for the 211 diffraction peak. The reconstructed and measured data differences were divided by the measured lattice spacing to normalize natural differences between MS and LTT and point-to-point differences due to residual stresses. Figure 3 summarizes how the average deviation converges for reconstructed strain fields from GPRs incorporating a greater fraction of the total measurement points. The average deviation converges to near zero for GPRs, which includes information from >45% of the total data points. Estimated errors in the reconstructed data of GPRs with >50% of the total data points match or exceeds the experimental uncertainty. Including >60% of the data points does not significantly decrease the variability in the average deviation. The convergence to a zero nominal deviation and estimated errors evidence that the GPR is robust and can estimate a data point using information from neighbors. While both samples converge to an estimated error that matches the experiment, LTT has a higher variability.

Calculating elastic strains and, subsequently, residual stresses can magnify errors in the interatomic spacing. The average absolute deviation between the reconstructed and measured elastic strains and stresses were investigated to determine if (1) recovered points capture the measured strain state and (2) fusing GPR strain fields introduces errors. Figure 4 shows how the fraction of data points included in the GPR affects the average deviation between the measured and reconstructed elastic strains and stresses. These data highlight that GPRs incorporating more information better reproduce the experimental strain fields. Variations in the magnitudes of the elastic strains are observed for the different strain directions, highlighting the underlying variability of the principal strains shown in Figure 2. Similar to Figure 3, observed deviations are higher in LTT than in Mild steel, and the variations are minimally reduced after incorporating 60% of the non-masked data into the GPR. Appendix A shows the change in average deviation with increased infraction of incorporated information.

The deviations in the von Mises stress were assessed as a tertiary check [15,16]. The von Mises equivalent stress, or von Mises stress criterion, was selected because this criterion represents the 3D stress state of the material as a scalar using the following relation:(5)$$\sigma_{v}=\sqrt{\frac{1}{2}\left[{\left(\sigma_{11}-\sigma_{22}\right)}^{2}+{\left(\sigma_{22}-\sigma_{33}\right)}^{2}+{\left(\sigma_{11}-\sigma_{33}\right)}^{2}\right]}$$

Similar to deviations in the elastic strains and stresses, the von Mises stress criterion also converges to a steady state after incorporating 60% of the measured data. The convergence of these quantities suggests that the GPR model has included sufficient information needed to reconstruct the underlying strain fields, and further increasing the experimental information does not affect the quality of the reconstruction. We note that determining an average absolute deviation can never approach 0, and the subsequent offset indicates the underlying variability in the measured data.

### 3.3. Effect of Initial Sampling on GPR Convergence

GPRs require an initialization sample for selecting what measurement would provide the maximum information. Two controlling parameters are available for defining the initialization: (1) population size and (2) sampling approach (e.g., random vs. structured grid). For simplicity, the 2D map of the scalar von Mises stress is used to understand how these two settings affect the reconstruction. Convergence was assessed using the structural similarity criteria (SSIM), a common metric used in the image processing literature because it emphasizes structures and is not impacted by local shifts in intensity values. Figure 5 compares the change in structural similarity for differing initial sample populations. The initial sample size has two noticeable effects on the evolution of SSIM with an increasing fraction of incorporated data. First, increasing the size of the initial sample has an overall impact on improving the initial structural similarity. Second, the increase in initial structural similarity comes at the expense that the higher initial fraction requires a higher sampled fraction to reach a saturated SSIM, as approximated by the region where increasing data does not significantly improve SSIM. More continuous stress fields in MS reach saturation at a lower sampled fraction than LTT.

Effects of a structured vs. random grid of initialization points on the change in SSIM with increasing fraction sampled is most prominent below ~30 and 60% for mild steel and LTT, respectively. Structured grids tend to have poorer SSIM in mild steel, while the effects are less pronounced in LTT, where there is no clear trend. The lack of a clear trend with LTT might originate from the presence of the underlying strain gradients. A randomized sampling approach could probabilistically initialize the GPR with points that do not sufficiently represent the strain distribution and require a higher sampled fraction to approach convergence. While there are caveats, ultimately, both samplers incorporate sufficient information that their reconstructed von Mises stress distributions are equivalent.

### 3.4. Influence of Mapping Resolution on GPR Convergence

Significant changes in the built-in lattice strain over a short distance can cause problems for strain mapping. Generally, a finer spacing between grid points can help to resolve more severe gradients in the elastic lattice strain. However, a finer measurement spacing can significantly increase the measurement time needed to sample a map. For example, reducing the grid from a 5 × 5 mm to 2.5 × 2.5 mm quadruples the number of points and time. A priori insight into the location of the gradients would allow researchers to intelligently define a measurement strategy using a multi-step mesh where some regions of interest have a higher density of measurement points. A GPR-based dynamic sampling approach offers greater flexibility because measured data from prior samples are used to identify areas of interest. Figure 6 compares the convergence in SSIM for mild steel and LLT data sampled using a 2.5 × 2.5 mm (fine) or 5 × 5 mm (coarse) grid of measurement points. Grid size had limited effects on the convergence of the mild steel reconstruction. However, the finer grid does not significantly impact the reconstruction of the LTT strain map, where the fine grid has a considerably better SSIM at a low sampled fraction than the coarse grid. Differences in the convergence are attributed to the high strain gradients near the base plate that are more difficult to reconstruct using the coarser grid.

## 4. Discussion

Differences between measured and masked/predicted locations suggest that LTT systematically has a higher error (Figure 3 and Figure 4). We attribute the apparent systematic error to the underlying differences in the strain patterning, where data near the high strain gradients can contain more information needed to reconstruct neighboring data. Beyond differences in LTT and MS, the key takeaway is that the GP approach properly recovers masked data using information from neighboring points. Data were recovered to an error below experimental uncertainties, as determined by error propagation.

The fraction of points used to initialize the GP and the method used to select the data overall had a minimal impact on the overall convergence. Observed differences were localized to the GPR that incorporated <40% of the total dataset. Structured and random initialization grids do not affect the overall convergence. All observed differences in the evolution of SSIM with increasing fraction for the randomly initialized data are attributed to the underlying random sampling, where the inherent variability of the randomization can affect the initialization. This study used different random points to mimic a dynamic experiment’s function. Increasing the initialization fraction tended not to change the evolution in SSIM, except that a 20% fraction qualitatively appeared to converge slightly more slowly. The lack of apparent effects of the initialization approach suggests that the present method is robust and that a low density of initialization data is sufficient to obtain accurate reconstructions. The implemented GP and Bayesian optimization methods minimize total uncertainties through exploration by selecting what point has the highest likelihood of improving the reconstruction. Strains in materials are continuous and likely contribute to the robustness of the reconstructions because these data should not have point-to-point discontinuities without the presence of artifacts. We note that artifacts can arise from various sources of errors, such as an artificial shift of gauge volume, e.g., partially filled gauge volume, heterogeneous microstructure, voids, etc.

As a comparison, Figure 7 shows the measured and reconstructed von Mises stress fields to visually show that data obtained from three independent GPRs capture the underlying distributions of the measured data. The difference between the measured and reconstructed von Mises stress was compared with the estimated experimental uncertainty to check if the GPRs correctly capture the stress magnitudes. For GPRs incorporating 60% of the possible measurement locations, data shown in Figure 7, the reconstructed von Mises stress is within the measurement uncertainty of 90% and 93% for MS and LTT, respectively. Locations that exceed the expected uncertainty are localized. Further investigation is needed to determine if the underlying localization of higher uncertainty deviations is evidence of an underlying experimental/numerical artifact or statistical variation. For example, each data point has a normal stress value with a representative uncertainty. The current approach does not account for the measured and reconstructed data overlap error bounds. The individual samples were on the uncertainty tails’ extreme positive and negative edges. Appendix A shows the difference maps of MS and LTT for various maximum fractions of incorporated points.

Effects of the strain gradients on how the GP and Bayesian optimization select points are highlighted in Figure 6, where the selection operation identifies a higher density near the strain gradients in LTT while a more uniform grid evolves for MS. Sample regions with a high density of points indicate that these regions have a higher degree of uncertainty. One factor affecting the uncertainty in this region is the large gauge volume used to measure these samples. The large gauge volume in this work was selected to decrease the time needed to achieve a sufficiently well-defined diffraction peak (~45 s/point). However, the larger gauge volume has the unintended consequence of smearing the gradient. This result also highlights the advantage of the GP-based approach, where a researcher could connect the GP as part of a library for dynamically driving experiments. Venkatakrishnan et al. [8] demonstrated the use of GP to dynamically sample a strain field using the GPR to identify the next measurement location. A higher-level implementation of this approach could add additional control to dynamically adjust the measurement gauge volume as needed to reduce the uncertainty of the strain map. Utilizing the method shown here offers significant advantages for defining non-standard complex grids that best represent a material’s underlying strain/stress fields.

Beyond locally changing the measurement GV, an implemented dynamic approach could change the specifics of the grid within identified regions of interest. For example, decreasing the mapping size from 5 × 5 mm to 2.5 × 2.5 mm over an entire strain map increases the total number of measurements by 4×. Many of these points capture minimal or no differences between neighbors. A more complex selection could implement a cost function to add constraints that could reduce the number of data points needed to reconstruct strain distributions. For example, neighboring points along sz provide more significant insights into strain gradients than those along sx (see Figure 2 or Figure 6).

## 5. Considerations

The results presented here demonstrated for an LTT and MS WAAM walls suggest that a GPR approach for dynamic sampling is viable for reconstructing the residual lattice strain distributions from a subset of measurement locations. Further, these reconstructed strain fields capture a high fidelity of information. The determined von Mises stress criteria highlight that the presented approach is valid for reconstructing a material’s 3D stress state using a subset of data, where >90% of the reconstructed data were within the estimated experiment uncertainty. Reconstructing complete strain fields from a subset of measurement locations offers to expand the measurement capabilities of strain mapping neutron diffraction instruments by enabling a finer GV to map components with a greater resolution or by reducing the time needed to complete measurements.

However, this approach is not without limitations, as it relies on a model that might not capture the underlying characteristics of the material. This work used previously measured data as the ground truth to assess if a GPR has reached a convergence. Deployment of this approach for real-world use will require a rigorous check for convergence to ensure that strain fields are reconstructed using a sufficiently high number of measurements. As evidenced in Figure 5 and Figure 6, the number of measured locations needed to achieve convergence can depend on the complexity of the underlying strain fields. One possible approach is incorporating finite element modeling results to check for convergence.

## Figures and Tables

**Figure 1 materials-16-03854-f001:**
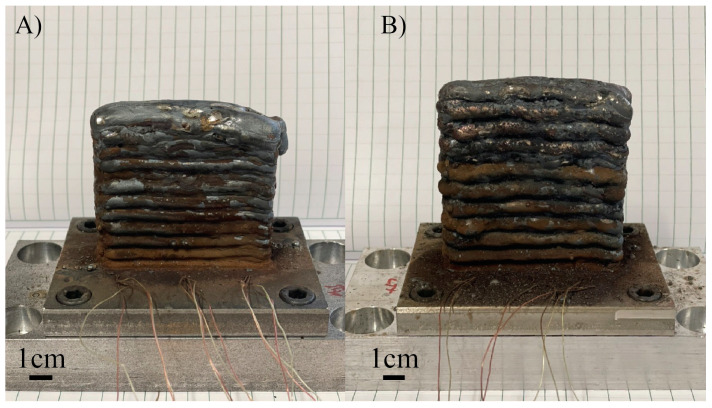
Picture of MS (**A**) and LTT (**B**) WAAM walls. Differences in the layer heights of each material affected the resultant build heights.

**Figure 2 materials-16-03854-f002:**
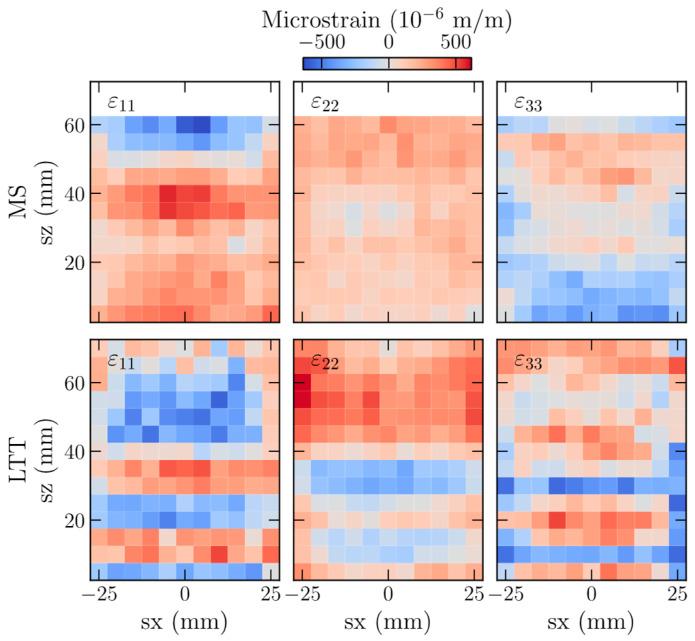
Comparison of residual strain maps measured along the 11, 22, and 33 principal strain directions for MS (**top**) and LTT (**bottom**). Differences in the material properties of mild steel and LTT give rise to disparate strain distributions, with LTT having sharp strain gradients near the base plate.

**Figure 3 materials-16-03854-f003:**
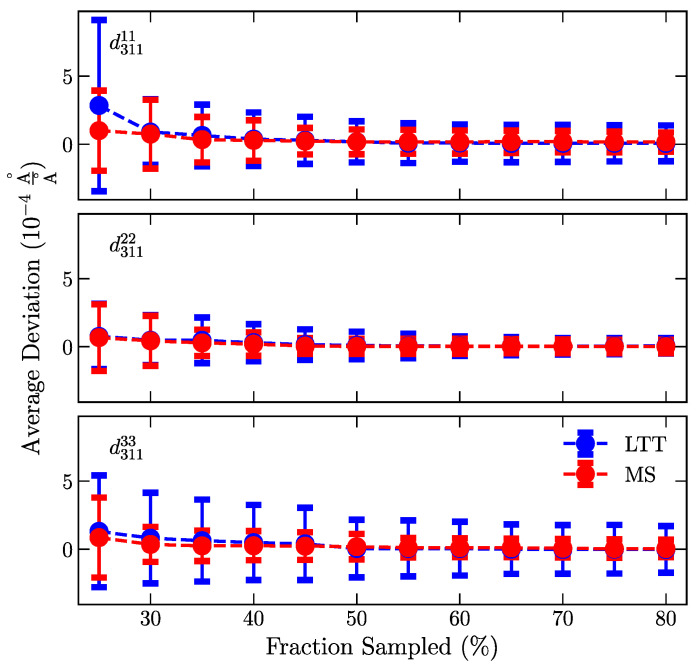
Converges of the average deviation between the reconstructed and measured interatomic spacing for mild steel (red) and LTT (blue). Increasing the fraction of the full set of map locations in the GPR reconstruction decreases the average deviation. Errors in the average deviation represent the standard deviation of all trials. The shaded region represents the experimental errors, as determined by the propagation of errors from peak fitting.

**Figure 4 materials-16-03854-f004:**
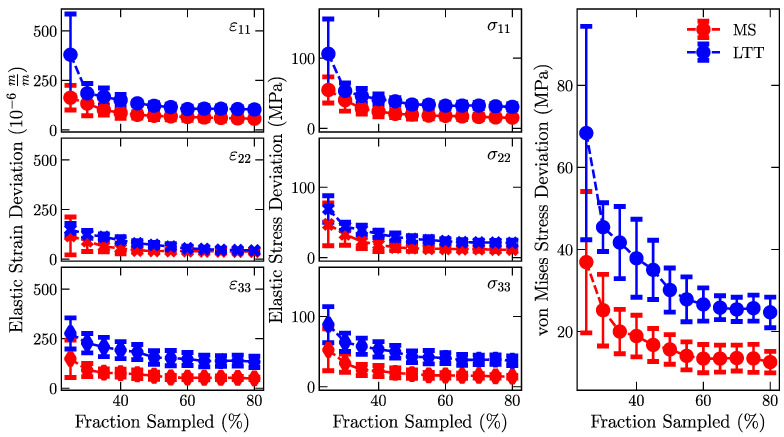
Comparison of how deviations in the estimated interatomic spacing of masked points propagate to errors in strain and, subsequently, stress. Data represent the average absolute deviation between the reconstructed and measured strains and stresses. Predicted data converge to a low variation for GPs that incorporate >50% of the experimental data. The higher deviations in LTT are attributed to the more significant strain gradients. Errors are assessed using the standard deviation in the average deviations for the ten trials.

**Figure 5 materials-16-03854-f005:**
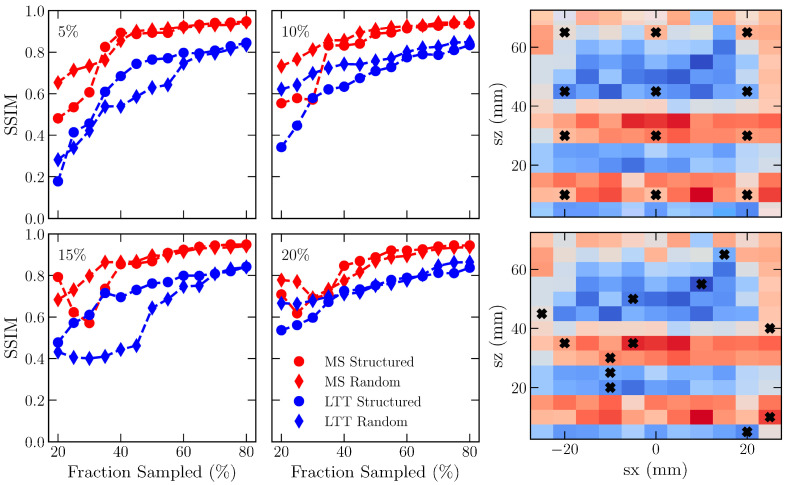
Convergence behavior of the SSIM between experimental and reconstructed von Mises stress maps for GPRs initialized with a 5, 10, 15, or 20% sampling fraction using either a structured or random point selection. Representative stress maps are shown with an overlay of the grid or random point selection to visualize differences in the initial sample.

**Figure 6 materials-16-03854-f006:**
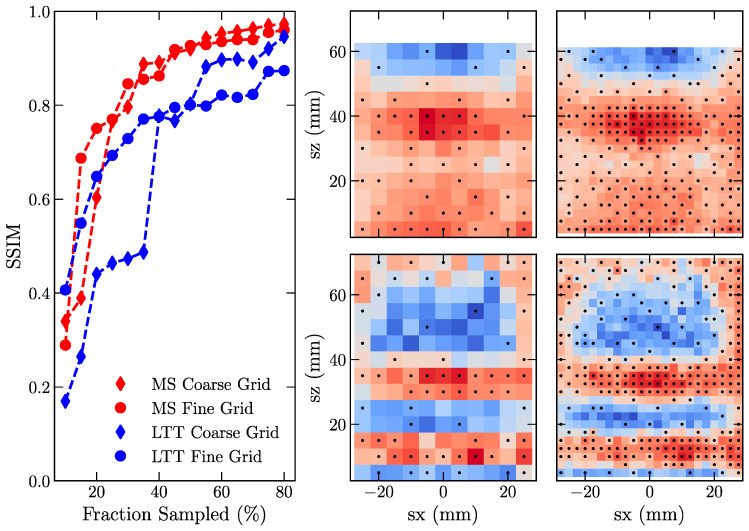
Change in structural similarity for reconstructed strain fields using a 5 mm (coarse) or 2.5 mm (fine) measurement. More prominent strain gradients in LTT affect convergence. Experimental strain maps are shown with overlays of points for reconstructions from GPRs that incorporate 50% of the measured data.

**Figure 7 materials-16-03854-f007:**
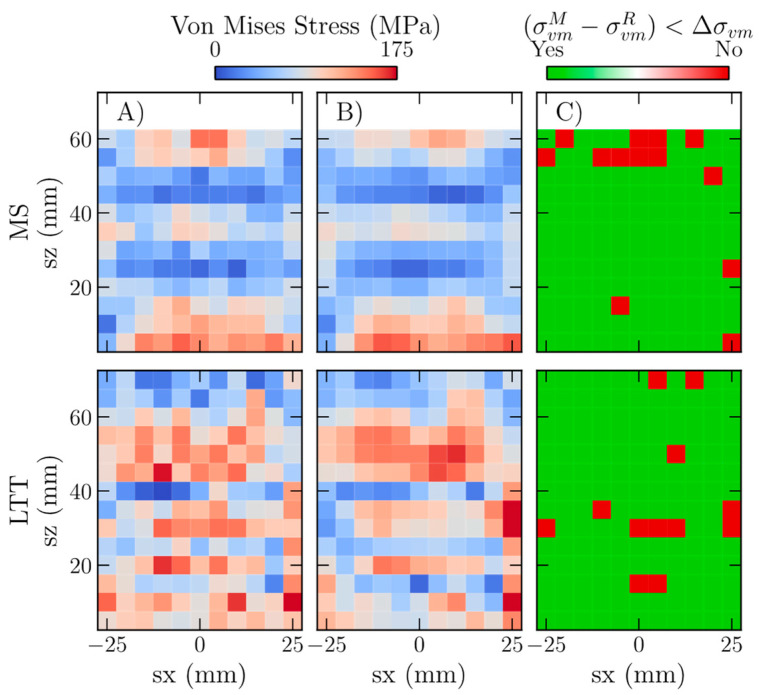
Comparison of the measured (**A**) and GPR reconstructed (**B**) von Mises stress fields with a map of locations where the difference between the measured and reconstructed stresses are within the experimental uncertainty (**C**). GPR reconstructions incorporated a 60% sampling fraction for MS and LTT. The difference map highlights that both reconstructions have a similar underlying structure and magnitudes of the von Mises stress field, as evidenced by 90% and 93% of the reconstructed data being within the estimated error in the measured data for MS and LTT, respectively. GPRs were initialized using a 10% random sample.

**Table 1 materials-16-03854-t001:** Summary of AM process setting used to fabricate MS and LLT planar walls.

Parameter	Mild Steel	LTT 250
Power	0.9 kW	1.1 kW
Wire Feed Rate	1905 mm/min	2794 min/min
Traverse Feed Rate	80 mm/min	80 mm/min
Argon Shielding Gas Flow Rate	15 cfm	15 cfm

## Data Availability

The data presented in this study are available on request from the corresponding author.

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
