# Peer review of "Validating the Use of Gaussian Process Regression for Adaptive Mapping of Residual Stress Fields"

_materials, 2023, doi:10.3390/ma16103854_

Round 1
Reviewer 1 Report
The research is interesting and very useful for improving testing efficiency. If the comparison of finite element simulation can be added, the effect will be better.
Author Response
The research is interesting and very useful for improving testing efficiency. If the comparison of finite element simulation can be added, the effect will be better.
Response: We thank the reviewer for their comments.
Reviewer 2 Report
The manuscript "Validating the use of Gaussian Process Regression for Adaptive Mapping of Residual Stress Fields” proposed Gaussian process regression to reconstruct 3D stress state with measured neutron residual strain maps. It is an interesting and meaningful idea to obtain the residual stress field with much less measurements. The manuscript is well written and I highly recommend it published in this journal. Some detailed comments can be found below.
1) The information of correspondence is missing.
2) Line 53, please clarify why 231 measurements are needed if 50*100 is sampled with 5*5 grid.
3) The information of Ref [8] is incomplete. I would like to read it first to check the difference between this work, but I failed to find it.
4) Initial sampling is very important for the GPR. Please give more detailed information about the sampling approach with random and structured grids. For instance, what is the mathematical function of the random grid and how it changes when the random grids with the same sampling fraction are applied?
5) After the discussions of initial sampling and mapping resolution on GPR convergence, it would be benefited for the readers if the comparison between the reconstructed stress field with the optimal parameters and the actual stress field can be provided.
6) Please check some text errors in the manuscript. For instance, “two-dimensional” should be “two-dimension” or “two dimensional”; Equation number should be (1), (2) et al.
Author Response
See attached document.

Reviewer 3 Report
Manuscript review „ Validating the use of Gaussian Process Regression for Adaptive Mapping of Residual Stress Fields”.
The manuscript is very interesting, but requires substantive and editorial correction.
The summary needs to be edited again. The abstract describes the content of the manuscript. parts of the text (17-21) should not be included in the summary.
Abstract - it should be redrafted, with no content that could be included in the very introduction to the manuscript
The summary should essentially cover the content of the manuscript with the explicit purpose of the study, which is unfortunately lacking in this manuscript.
In the introduction to the manuscript, there is no clear indication of the practical reason for stress analysis using the method indicated by the authors.
There is no indication of the practical (application) application of the indicated method - where and in what cases is this method better in comparison to other methods used in practice?
The described methodology lacks a clear indication of the research plan.
There is no indication of the number of completed experiments and the number of tested samples.
The lack of this information affects the credibility and repeatability of the presented studies.
The authors did not present any statistical analysis (referring to all samples), which is a must in the case of experimental research.
Yes, such an analysis was carried out, but only in relation to the mapping of residual deformations on a single selected sample (this is to be expected) - fig.2; Fig.3; Fig.4.
It is very important to describe the subject of research correctly and in detail - detailed characteristics of the research method itself, but also a detailed description of a single sample (subject of research) - this should be completed.
For example, in the process of additive application of successive layers, it is very important to determine the thickness of these layers, which can range from 0.02 - 0.08 mm.
It is the thickness of the successively applied layers that determines, among others, the correct selection of laser power and the value of stresses. Complete the parameters described in table 1 (99)
Unfortunately, the authors did not refer to the influence of the method of preparing samples for testing.
Even the preparation itself (e.g. initial surface treatment) can have a significant impact on the stress value in the product.
Also, the authors do not indicate the significance of the arrangement (position) of the analyzed (analyzed) samples.
The authors indicate the influence of geometric, thermal and material parameters on homogeneity, chemical composition and microstructure. It would be good to supplement the manuscript with examples of structures with a description of their components.
There are no conclusions regarding the impact of the technology on the stress value and mechanical properties.
Conclusions are very cursory and can certainly be supplemented - for example, with an analysis of the impact of the use of the technology itself on the value of stresses and the value of other mechanical properties - for example Re(Rp0.2) , Rm, A5; HB (HRB) etc.
Basically the text of the manuscript very interesting, but needs to be revised.
Author Response
The manuscript is very interesting, but requires substantive and editorial correction.
The summary needs to be edited again. The abstract describes the content of the manuscript. parts of the text (17-21) should not be included in the summary.
Response: The abstract has been updated.
Abstract - it should be redrafted, with no content that could be included in the very introduction to the manuscript
Response: The abstract has been updated.
The summary should essentially cover the content of the manuscript with the explicit purpose of the study, which is unfortunately lacking in this manuscript.
Response: We thank the reviewer for their comment. In this work, we utilize a “considerations” instead of a traditional conclusion to give the reader context about how the method.
In the introduction to the manuscript, there is no clear indication of the practical reason for stress analysis using the method indicated by the authors.
Response: The goal of the study was to determine if a GPR approach is viable for reducing number of data needed to determine the 2D stress state of materials, using a neutron strain mapping instrument. We have more clearly delineated this goal in the introduction “In this work, we determine the feasibility of using the GPR approach to reduce the number of independent measurements needed to reconstruct the 2D stress distributions in WAAM walls. Uncertainty in the reconstructed strain fields was assessed by masking points from the GPR sample and determining the difference between the recovered and measured data. Effects of how these differences propagate to uncertainties in the resulting residual stresses are explored. Results presented evidence that calculated residual stresses using a reconstructed strain field is valid, and resulting stresses are in excellent agreement with stresses calculated from full strain maps, demonstrating that a GPR based autonomous experiment is a feasible approach for reducing the number of data points needed to capture the stress state of a material.”
There is no indication of the practical (application) application of the indicated method - where and in what cases is this method better in comparison to other methods used in practice?
Response: See above comment
The described methodology lacks a clear indication of the research plan.
Response: We refer the reviewer to “The overall performance of the GPR model was quantified using a modified K-fold cross-validation procedure. A random set of points (10%) were masked and excluded from the GPR training set, and the performance of the GPR was assessed based on the predictions for these points compared against the withheld data. For consistency, masked points were held constant for all strain directions. Differences between the measured and recovered data for the masked locations were calculated. These differences were evaluated for raw interatomic spacing, lattice strain, and residual stress. A set of ten trials ( ) was used to test the error variability. Each GPR was initialized using 16, and 14 randomly selected non-masked locations for the LTT and MS, respectively – about 10% of the total grid size. Effects of the initialization approach were assessed by comparing how structured vs. random initial point selection strategies and the fraction of initialization points included impacted the convergence of the reconstructed strain maps. Structured grids were devised to select points that are evenly spaced throughout the wall. The random initialization grid was constructed by iteratively selecting pixel ids from a list of non-masked pixel ids via the numpy random.randint function. Selected pixel ids were removed from the selection list to ensure a pixel id could only be selected once. Pixel ids were selected until sufficient points were identified to achieve the desired initialization fraction. Figure S1 visualizes three random initialization grids to illustrate how the grids differ.”
There is no indication of the number of completed experiments and the number of tested samples.
Response: The aim of this work was to determine if a GPR approach was viable for reconstructing residual stress fields in wire-arc additively manufactured walls. This approach was applied to two walls with different chemistries.
The lack of this information affects the credibility and repeatability of the presented studies.
The authors did not present any statistical analysis (referring to all samples), which is a must in the case of experimental research.
Response: Additional details of the samples studied in this work have been added, including pictures of the fabricated walls.
Yes, such an analysis was carried out, but only in relation to the mapping of residual deformations on a single selected sample (this is to be expected) - fig.2; Fig.3; Fig.4.
Response: Additional details of the samples studied in this work have been added, including pictures of the fabricated walls.
It is very important to describe the subject of research correctly and in detail - detailed characteristics of the research method itself, but also a detailed description of a single sample (subject of research) - this should be completed.
Response: Additional details of the samples studied in this work have been added, including pictures of the fabricated walls.
For example, in the process of additive application of successive layers, it is very important to determine the thickness of these layers, which can range from 0.02 - 0.08 mm.
Response: This work fabricated samples using a wire-arc additive approach with layer thickness >1mm
It is the thickness of the successively applied layers that determines, among others, the correct selection of laser power and the value of stresses. Complete the parameters described in table 1 (99)
Response: Parameters describe in table 1 are a complete representation of what are used for WAAM.
Unfortunately, the authors did not refer to the influence of the method of preparing samples for testing.
Response: Samples were measured in the as-fabricated conditions and had not been removed from the baseplates. These details have been included in the methods.
Even the preparation itself (e.g. initial surface treatment) can have a significant impact on the stress value in the product.
Response: Samples were measured in the as-fabricated conditions and no surface sample preparation was used.
Also, the authors do not indicate the significance of the arrangement (position) of the analyzed (analyzed) samples.
Response: Diffraction data were measured for 3 orthogonal strain directions as is standard practice for neutron strain mapping experiments.
The authors indicate the influence of geometric, thermal and material parameters on homogeneity, chemical composition and microstructure. It would be good to supplement the manuscript with examples of structures with a description of their components.
Response: Additional details of the samples studied in this work have been added, including pictures of the fabricated walls used in this work.
There are no conclusions regarding the impact of the technology on the stress value and mechanical properties.
Response: This work is focused on validating the use of a ML approach for reducing the number of measurements needed to reconstruct strain and subsequent stress fields in WAAM walls.
Conclusions are very cursory and can certainly be supplemented - for example, with an analysis of the impact of the use of the technology itself on the value of stresses and the value of other mechanical properties - for example Re(Rp0.2) , Rm, A5; HB (HRB) etc.
Response: This work is focused on validating the use of a ML approach for reducing the number of measurements needed to reconstruct strain and subsequent stress fields in WAAM walls.
Round 2
Reviewer 3 Report
Unfortunately, I don't see that most of my comments have been taken into account.
The changes made are minimal.
Manuscrypt is a description, a research report and should be corrected according to the comments in the first review.
Author Response
A response to the prior review is provided in a word document
